# Venous Thromboembolism and Primary Thromboprophylaxis in Perioperative Pancreatic Cancer Care

**DOI:** 10.3390/cancers15143546

**Published:** 2023-07-08

**Authors:** R. A. L. Willems, N. Michiels, V. R. Lanting, S. Bouwense, B. L. J. van den Broek, M. Graus, F. A. Klok, B. Groot Koerkamp, B. de Laat, M. Roest, J. W. Wilmink, N. van Es, J. S. D. Mieog, H. ten Cate, J. de Vos-Geelen

**Affiliations:** 1Department of Functional Coagulation, Synapse Research Institute, 6217 KD Maastricht, The Netherlands; ruth.willems@mumc.nl (R.A.L.W.); b.delaat@thrombin.com (B.d.L.); 2Thrombosis Expert Center Maastricht, Maastricht University Medical Center, 6202 AZ Maastricht, The Netherlands; h.tencate@maastrichtuniversity.nl; 3Department of Internal Medicine, Section Vascular Medicine, Maastricht University Medical Center, 6202 AZ Maastricht, The Netherlands; 4Department of Internal Medicine, Section Medical Oncology, Maastricht University Medical Center, 6202 AZ Maastricht, The Netherlands; merlijn.graus@mumc.nl; 5CARIM, School for Cardiovascular Diseases, 6229 ER Maastricht, The Netherlands; 6Department of Surgery, Leiden University Medical Center, 2333 ZA Leiden, The Netherlands; n.michiels@lumc.nl (N.M.); j.s.d.mieog@lumc.nl (J.S.D.M.); 7Department of Internal Medicine, Section Vascular Medicine, University of Amsterdam, Amsterdam UMC Location, 1105 AZ Amsterdam, The Netherlands; v.r.lanting@amsterdamumc.nl (V.R.L.); n.vanes@amsterdamumc.nl (N.v.E.); 8Amsterdam Cardiovascular Sciences, Pulmonary Hypertension and Thrombosis, 1081 HV Amsterdam, The Netherlands; 9Tergooi Hospitals, Internal Medicine, 1201 DA Hilversum, The Netherlands; 10Department of Surgery, Maastricht University Medical Center, 6202 AZ Maastricht, The Netherlands; stefan.bouwense@mumc.nl; 11NUTRIM, Maastricht University Medical Center, 6229 ER Maastricht, The Netherlands; 12Department of Surgery, Erasmus University Medical Center, 3015 CN Rotterdam, The Netherlands; b.vandenbroek@erasmusmc.nl (B.L.J.v.d.B.);; 13GROW, Maastricht University Medical Center, 6229 ER Maastricht, The Netherlands; 14Department of Medicine-Thrombosis and Hemostasis, Leiden University Medical Center, 2333 ZA Leiden, The Netherlands; f.a.klok@lumc.nl; 15Department of Platelet Pathophysiology, Synapse Research Institute, 6217 KD Maastricht, The Netherlands; m.roest@thrombin.com; 16Cancer Center Amsterdam, 1081 HV Amsterdam, The Netherlands; j.wilmink@amsterdamumc.nl; 17Department of Medical Oncology, Amsterdam University Medical Center, Location University of Amsterdam, 1105 AZ Amsterdam, The Netherlands

**Keywords:** pancreatic ductal adenocarcinoma, cancer-associated thrombosis, venous thromboembolism, neoadjuvant treatment, chemotherapy, anticoagulation therapy

## Abstract

**Simple Summary:**

Historically, patients with pancreatic ductal adenoma carcinoma were subjected to immediate surgical resection of the pancreatic tumor. Nowadays, more and more patients are treated with chemo(radio)therapy before surgical resection. It is known that patients with pancreatic cancer have a high risk of developing thrombosis. However, as patients underwent immediate surgery before, the incidence of thrombosis in patients with pancreatic cancer during neoadjuvant chemotherapy is understudied. Few studies have investigated the VTE incidence in this population and it is unclear whether these patients should use perioperative thromboprophylaxis to prevent thrombosis. This narrative review summarizes the evidence that is currently available.

**Abstract:**

Recent studies have shown that patients with pancreatic ductal adenocarcinoma (PDAC) treated with neoadjuvant chemo(radio)therapy followed by surgery have an improved outcome compared to patients treated with upfront surgery. Hence, patients with PDAC are more and more frequently treated with chemotherapy in the neoadjuvant setting. PDAC patients are at a high risk of developing venous thromboembolism (VTE), which is associated with decreased survival rates. As patients with PDAC were historically offered immediate surgical resection, data on VTE incidence and associated preoperative risk factors are scarce. Current guidelines recommend primary prophylactic anticoagulation in selected groups of patients with advanced PDAC. However, recommendations for patients with (borderline) resectable PDAC treated with chemotherapy in the neoadjuvant setting are lacking. Nevertheless, the prevention of complications is crucial to maintain the best possible condition for surgery. This narrative review summarizes current literature on VTE incidence, associated risk factors, risk assessment tools, and primary thromboprophylaxis in PDAC patients treated with neoadjuvant chemo(radio)therapy.

## 1. Introduction

Pancreatic ductal adenocarcinoma (PDAC) is a lethal form of cancer. PDAC is the fourth leading cause of cancer-related death in the United States and Europe [1,2]. About 20% of patients with PDAC are eligible for surgical resection at the time of diagnosis, the only potentially curative treatment [3].

Historically, patients with resectable PDAC underwent surgery upon diagnosis. The PREOPANC trial has demonstrated that neoadjuvant chemoradiotherapy in patients with resectable and borderline resectable pancreatic cancer improves overall survival (OS) when compared to upfront surgery [4]. Currently, several neoadjuvant chemotherapy regimens are being studied [5,6,7,8,9,10,11]. Following these trials, an increasing number of patients with (borderline) resectable and locally advanced PDAC are treated with chemotherapy before surgery.

Venous thromboembolism (VTE) is a leading cause of death in cancer patients, following cancer progression [12]. Among all cancer types, pancreatic cancer is associated with the highest risk of developing VTE [13]. Approximately one in five patients with newly diagnosed pancreatic cancer develop VTE [14,15,16,17,18,19,20,21]. VTE development is associated with shorter survival times [22].

Current guidelines suggest or recommend thromboprophylaxis for ambulatory patients with locally advanced and metastatic PDAC treated with chemotherapy [23,24]. Notably, no specific guideline recommendations are available for ambulatory patients with localized PDAC. However, it is important to prevent complications during neoadjuvant treatment to keep these patients in the best possible condition prior to surgery. 

This narrative review focuses on VTE incidence in patients with potentially resectable PDAC undergoing neoadjuvant chemotherapy, and addresses the effects of primary thromboprophylaxis in the neoadjuvant trajectory. 

## 2. Incidence of VTE in Patients with PDAC

The reported incidence of VTE in patients with metastatic PDAC varies from 12 to 36% [14,15,16,17,18,19,20,21]. Prior to the trials showing a survival benefit for patients treated with neoadjuvant chemotherapy, these patients underwent surgery immediately upon diagnosis [4,25,26].

The literature on the preoperative incidence of VTE in patients with (borderline) resectable PDAC is limited (Table 1). The trials studying different neoadjuvant approaches for patients with (borderline) resectable PDAC did not report data on VTE incidences [4,5,6,7,8,9,10]. The available literature shows preoperative VTE incidences ranging from 11 to 14% for patients with borderline resectable PDAC, and 8–21% for resectable PDAC, treated with different neoadjuvant chemotherapy regimens.

Notably, the study performed by Krepline et al. observed a lower rate of neoadjuvant therapy completion and subsequent surgery in patients who developed VTE (54%) compared to patients who did not develop VTE (75%) (OR = 0.38; 95% CI [0.17–0.86], *p* = 0.02). In addition, the median OS was found to be decreased, although not significant, in patients who developed VTE (17 months versus 25 months, *p* = 0.11). This potential association between VTE incidence and survival could be due to VTE-related complications, including fatal PE, anticoagulant-related bleeding, or a delay of cancer treatment. The association could also reflect an epiphenomenon of cancer progression, since VTE tend to occur more frequently in patients with biologically aggressive tumors or in those with progressive disease.

In conclusion, data on VTE incidence in patients with localized PDAC prior to surgery are scarce and the potential additional risk of chemotherapy on VTE development in this patient group is understudied.

## 3. Risk Factors of PDAC-Associated Thrombosis

The development of thrombosis in patients with cancer is a complex and multifactorial process involving various risk factors related to the individual patient, cancer, and cancer treatment.

### 3.1. General Individual Risk Factors

As in the general population, individual risk factors increase the risk of developing thrombosis in patients with cancer [33,34]. A well-known thrombosis risk factor in the general population is age [32]. The increase in VTE risk with age is also observed in patients with cancer [32,35,36]. Female sex is a risk factor for VTE in patients with PDAC, as well as in other cancer types [22,32,37]. Moreover, several comorbidities are associated with VTE in patients with cancer, such as obesity (OR = 1.52), pulmonary disease (OR = 1.37), renal disease (OR = 1.53), infection (OR = 1.77), and anemia (OR = 1.35) [36,38]. Cancer patients who have a history of VTE have a 6-fold higher risk of developing VTE [39].

A major risk factor for VTE in the general population is hospitalization [40]. VTE incidence in hospitalized patients with cancer is 8% and is associated with in-hospital mortality [41]. Immobility in cancer patients is assessed with performance status; in cancer patients with a poor performance status, higher VTE rates were observed [42,43]. Bed rest for longer than 3 days is also associated with VTE in patients with cancer [39].

### 3.2. Stage- and Tumor-Location-Related Risk Factors

The primary cancer site itself is shown to affect the risk of VTE. Gastrointestinal tumors have been associated with a higher risk of VTE development [44]. Patients with PDAC display the highest rate of VTE: approximately one in five patients develop thrombosis [14,15,16,17,18,19,20,21,45]. Over recent years, the incidence of VTE has been increasing in patients with PDAC [46]. Tumors in the cauda and corpus of the pancreas result in a 2–3-fold increased risk compared to tumors located in the caput [15,22]. Additionally, VTE is observed in patients with locally advanced and metastatic PDAC more often than in patients with (borderline) resectable PDAC [22,32,46]. The median time to VTE is about four months following PDAC diagnosis [22].

### 3.3. PDAC-Treatment-Related Risk Factors

Apart from the cancer itself, cancer treatment also seems to increase VTE risk. Treatment with antineoplastic agents has been associated with an increased VTE risk in patients with all-type cancer [47,48,49,50]. In patients with PDAC, chemotherapy as the first-line cancer treatment is associated with an increased VTE incidence [22,51]. Withinthe neoadjuvant setting, patients with borderline resectable and locally advanced PDAC are commonly treated with FOLFIRINOX chemotherapy or gemcitabine-based chemoradiation in borderline resectable PDAC [4,52,53]. Gemcitabine, oxaliplatin and 5-FU have been described to associate with thrombosis [54,55,55,56].

In patients with metastatic PDAC, no significant difference in thromboembolism incidence was found when comparing gemcitabine to FOLFIRINOX (4 vs. 7%) [57]. Data on VTE incidence have not been reported in the clinical trial comparing gemcitabine to nab-paclitaxel plus gemcitabine in patients with metastatic pancreatic cancer [58].

Chemotherapy may induce neutropenia, for which hematopoietic growth factors can be administered. In a prospective study of 731 patients with a new diagnosis of PDAC at any stage, the use of granulocyte colony-stimulating factors was associated with a higher VTE rate (HR = 1.66) [22].

Central venous catheter (CVC) use is common for the administration of chemotherapy. In patients with PDAC, CVC-related thrombosis incidences around 5% have been reported [22,32].

## 4. Selecting Patients at Risk of VTE

Biomarkers and risk assessment tools for VTE may help physicians in identifying patients with PDAC at the highest risk of developing VTE. By differentiating patients according to VTE risk, patients who are likely to benefit most from primary thromboprophylaxis can be selected. Targeted thromboprophylaxis is desirable in patients with cancer, considering the concurrent higher risk of bleeding [59].

### 4.1. Biomarkers for VTE Risk in PDAC

In patients with cancer, multiple studies have been performed to identify biomarkers predictive of cancer-associated thrombosis (CAT). Altered blood counts are often found in patients with cancer, especially in patients with gastrointestinal cancer, and are associated with VTE occurrence [60,61]. A hemoglobin level of <6.2 mmol/L is associated with a higher risk of VTE (OR = 1.8–2.4) [42,50]. Moreover, patients with cancer with high platelet counts have a higher rate of VTE when compared to patients with normal platelet counts (OR = 2.8–3.5) [42,62]. The biomarkers of platelet activation have also been studied in CAT [62,63]. PDAC patients with an elevated platelet factor 4 (PF4) levels before treatment have a 2.7-fold increased risk of VTE development [64]. In addition, leukocytosis has also been associated with a higher VTE risk in several studies (leukocyte count >11 × 10^9^/L, OR = 2.2–2.4) [50,62,65]. A hemoglobin level <6.2 mmol/L, platelet count ≥350 × 10^9^/L, and leukocyte count >11 × 10^9^/L are included as variables in the Khorana score (Table 2), a CAT risk score that is increasingly used to aid in the decision of starting thromboprophylaxis in all-type cancer patients [66].

Thrombin is a central enzyme in the coagulation cascade. Patients with cancer with an elevated peak thrombin generation level (≥611 nM) in plasma have an increased rate of VTE with a HR of 2.1 [72]. D-dimer levels reflect the formation and degradation of fibrin. In patients with cancer, increased D-dimer levels pretreatment are associated with an increased VTE risk [68,73,74,75,76]. Plasminogen activator inhibitor-1 (PAI-1) inhibits the activators of plasminogen and hence fibrinolysis [77]. Active PAI-1 levels are associated with an increased risk of VTE in patients with PDAC [61]. Another hemostatic marker associated with VTE risk in patients with PDAC is tissue factor activity, associated with circulating extracellular vesicles which are shed by pancreatic cancer cells [76,78,79,80,81,82].

The majority of these studies investigated CAT biomarkers in pooled patient populations with different types and stages of cancer. A retrospective cohort study specifically reviewed VTE occurrence in 426 patients with PDAC administered neoadjuvant treatment before surgical resection [83]. A pretreatment hemoglobin level below 6.2 mmol/L was an independent predictor of VTE during neoadjuvant treatment (OR = 6.54). Moreover, a pretreatment platelet count higher than 443 × 10^9^/mL was a significant predictor of VTE, with an OR of 5.63. Another study has shown that CA19.9 levels are higher in PDAC patients with VTE compared to patients without VTE and that CA19.9 levels increase with the extent of VTE [84]. Boone et al. showed a lower percentage decrease in CA19-9 levels in response to neoadjuvant therapy to be protective for VTE, which is likely due to low CA19.9 levels at baseline [83].

Currently, most hemostatic markers are analyzed in plasma, and therefore blood-cell-dependent effects are generally not measured. The SENEPANC study, a multicenter multination prospective cohort study is currently investigating the predictive value of novel hemostatic markers in patients with PDAC, including the thrombin generation assay in whole blood [85].

### 4.2. VTE Risk Assessment Tools in PDAC

Clinical guidelines recommend pharmacological thromboprophylaxis with either reduced-dose DOAC or prophylactic-dose LMWH for the prevention of VTE in ambulatory cancer patients judged to be at a high risk of VTE [24,86,87]. Several pan-cancer risk assessment tools are available to identify such patients, including the well-validated Khorana score [67], Vienna model [68], ONCOTHROMB score [69], and ONKOTEV score [71] (see Table 2 for an overview of risk assessment tools). In most of these scores, pancreatic cancer is classified as a high-risk or very-high-risk tumor type. Of note, most of these scores were derived and validated in cancer patients with advanced disease receiving palliative systemic treatment. As such, their performance in patients with localized disease, including pancreatic cancer patients, is unclear. In particular, the absolute VTE incidence during neoadjuvant chemo(radio)therapy in this group is likely to be lower than reported in the derivation and validation studies which usually follow patients for 6 months.

The Khorana score is the most widely validated tool for VTE risk assessment. When using the positivity threshold of 2 points, all patients with pancreatic cancer are considered to be at a high risk of VTE and are therefore eligible for thromboprophylaxis according to the current ASCO, ITAC, and NCCN guidelines [24,86,88]. As such, it is questionable whether this score could be applied in PDAC patients to identify patients with an even higher risk of VTE. Subgroup analyses regarding the performance of the Khorana score in PDAC patients have shown that the absolute 6-month incidence of VTE in patients with a high-risk Khorana score ranges from 12% (pulmonary embolism (PE) or deep venous thrombosis (DVT) only) to 41% (superficial vein thrombosis and abdominal VTE included); in one study, the Khorana score had an AUROC of 0.65 [19,89,90,91,92]. A recent large population-based cohort study showed that the Khorana score was unable to identify PDAC patients at a particularly high risk of VTE (HR = 1.03, 95% CI [0.66–1.61]) [66]. In contrast, the Khorana score identified patients at a two-fold increased risk in an individual patient data meta-analysis of randomized controlled trials [93]. Nonetheless, the performance of this score during neoadjuvant treatment is less clear.

Given the modest performance of the Khorana score, several new risk assessment tools have been introduced, which mostly extend the Khorana score with additional clinical items or biomarkers (Table 2). The Vienna model had a better c-statistic than the Khorana score in the derivation study, i.e., indicating a better predictive value [68]. The score was externally validated, but its performance in a subgroup of PDCA patients specifically was not evaluated [94]. ONKOTEV is a clinical score with an AUROC of 0.76 in the derivation study [71]. A retrospective external validation study including 165 PDAC patients showed that ONKOTEV was also able to discriminate between high- and low-risk patients in this group [95].

Recently, a new VTE risk score was proposed which combines clinical items with a genetic germline risk score, including prothrombotic variants, i.e., the ONCOTHROMB score [69]. In the derivation and validation cohort, the score appeared to have a better discrimination than the Khorana score (validated AUROC 0.69 versus 0.58), although it will not be widely used in current clinical practice due to the need for genetic testing. Data on the subgroup of PDAC patients were not reported. Another recently proposed score by Li et al. modifies and extends the Khorana score by reclassifying the tumor risk groups, as well as by adding clinical items [70]. In their large derivation (*n* = 9769) and validation cohorts (*n* = 79,517), this score also outperformed the Khorana score (validated AUROC 0.68 versus 0.60). It is still unclear whether these findings can be extrapolated to patients with localized PDAC. Therefore, new prospective studies and post hoc analyses of existing studies are needed to address this knowledge gap.

## 5. Primary Thromboprophylaxis in PDAC

Several trials have been performed evaluating the efficacy and safety of prophylactic anticoagulants to prevent VTE in patients with cancer treated with chemotherapy (Table 3).

The CONKO-004 trial and FRAGEM trials investigated thromboprophylaxis with a low-molecular-weight heparin (LMWH) in patients with locally advanced and metastatic PDAC [96,97]. The CONKO-004 trial studied a high prophylactic dose of enoxaparin, while the FRAGEM trial studied a weight-adjusted dose of dalteparin. Both trials found thromboprophylaxis to be effective in preventing VTE, without increasing the incidence of major bleeding.

**Table 3 cancers-15-03546-t003:** Thromboprophylaxis in ambulatory patients with PDAC treated with chemotherapy to prevent VTE.

	LMWH				DOAC	
**Study**	**CONKO-004** Pelzer et al. (2015) [97]	**FRAGEM** Maraveyas et al. (2012) [96]	**PROTECHT** Agnelli et al. (2009) [98]	**SAVE-ONCO** Agnelli et al. (2012) [99]	**AVERT** Carrier et al. (2019) [100,101]	**CASSINI** Vadhan-Raj et al. (2020) [102]
**Cancer stage**	LAPC or metastatic	LAPC, recurrent or metastatic	LAPC or metastatic	LAPC or metastatic	Newly diagnosed or progression, all cancer stages	All stages
**No metastases** **(I/C)**	26/22	52/41 LAPC: 31/26 Metastatic: 29/37	-	-	-	39/36 Stage I/II: 21/15 Stage III: 14/17 Stage IV: 61/65
**Chemotherapy regimen**	Gemcitabine or Gemcitabine+ 5-FU+Cisplatin	Gemcitabine	-	-	-	5-FU-based or gemcitabine-based or gemcitabine+ Capecitabine /5-FU
**Study** **intervention (I)**	Chemotherapy alone or chemotherapy plus enoxaparin 1 mg/kg once daily	Gemcitabine alone or gemcitabine plus dalteparin 200 IU/kg once daily for 4 weeks followed by 150 IU/kg once daily for 8 weeks	Nadroparin 3800 IU once daily or placebo	Semuloparin 20 mg once daily versus placebo	Apixaban 2.5 mg twice daily or placebo	Rivaroxaban 10 mg once daily or placebo
**Duration** **intervention**	3 months	12 weeks	Duration chemotherapy	Duration chemotherapy	180 days	180 days
**Follow-up**	3 months	100 days	Duration intervention plus 10 days	Duration intervention plus 3 days	210 days	180 days
**VTE (I/C)**	1.3% vs. 10.2% *p* = 0.001 NNT = 11	3% vs. 23% *p* = 0.002 NNT = 6	5.9% vs. 8.3% NNT = 42	2.4% vs. 10.9% NNT = 12	5% vs. 16% *p* = 0.039 NNT = 9	3.7% vs. 10.1% NNT = 15
**MB (I/C)**	4.5% vs. 3.4% NS NNH = 76	3.2% vs. 3.4% NS	-	-	5% vs. 3% NS NNH = 50	1.5% vs. 2.3% NS NNH = 125

PROTECHT, SAVE-ONCO, AVERT, and CASSINI included patients with cancer, including a subset of patients with pancreatic cancer. Only the outcome data of patients with pancreatic cancer are reported in this table. 5-FU: 5-fluorouracil; DOAC: direct oral anticoagulant; I/C: intervention/control; LAPC: locally advanced pancreatic cancer; LMWH: low-molecular weight heparin; MB: Major bleeding; NNH: number needed to harm; NNT: number needed to treat; NS: not statistically significant; PC: pancreatic cancer; VTE: venous thromboembolic event.

The CASSINI and AVERT trials studied the efficacy and safety of a prophylactic dose of direct oral anticoagulants (DOACs) in ambulatory patients with cancer starting chemotherapy, with a Khorana score ≥2 (Table 3) [103]. Subgroup analyses were performed of the CASSINI trial in ambulatory patients with all stages of PDAC (*n* = 273), investigating the efficacy and safety of prophylactic rivaroxaban [102]. Rivaroxaban was effective in preventing DVT, PE, and VTE-related death (HR = 0.35, 95% CI [0.13–0.97]) in this subgroup analysis, whereas no difference in on-treatment major bleeding and clinically relevant non-major bleeding was observed compared to placebo [104]. Of the AVERT trial, studying a prophylactic dose of apixaban, a post hoc analysis was performed in patients with gastrointestinal cancers (*n* = 130) [100,101]. When comparing the apixaban to the control arm, a significant decrease in VTE incidence was found (HR = 0.45, 95% CI [0.21–0.96]), and there was no difference in major bleeding and clinically relevant non-major bleeding. Notably, all major bleeding events occurred in patients with pancreatic cancer.

Frere et al. performed a systematic review and meta-analysis of randomized controlled trials, evaluating the benefit of anticoagulants for primary VTE prevention in ambulatory patients with PDAC treated with chemotherapy [105]. Both studies investigating LMWHs and DOACs were included. This meta-analysis merged the data of the FRAGEM, CONKO-004, PROTECHT, SAVE-ONCO, and CASSINI trials. The study found the incidence rate of VTE to be lower in ambulatory PDAC patients treated with primary thromboprophylaxis (4%) compared to the placebo or no anticoagulant treatment (12%). The risk of VTE was decreased, with a pooled risk ratio (RR) of 0.31 and an estimated number needed to treat (NNT) of 12 patients. When comparing parenteral anticoagulants to oral anticoagulants, no difference in risk reduction was observed. When evaluating the safety, no difference was found in the incidence of major bleeding between patients receiving primary thromboprophylaxis and those receiving placebo or no treatment. The estimated number needed to harm (NNH) was 385 patients for major bleeding. No significant difference in the increased risk of major bleeding was found between parenteral and oral anticoagulants.

The results of the performed trials and meta-analysis suggest that thromboprophylaxis in ambulatory patients with PDAC treated with chemotherapy is effective and safe, and thus advisable. However, no trials have been performed that specifically study the efficacy and safety of thromboprophylaxis in patients with localized PDAC treated with chemotherapy in the neoadjuvant setting. Moreover, the chemotherapy regimens that were administered during the trials were heterogeneous and differ from the currently administered chemotherapy regimens, and patients at risk of bleeding were excluded. Further clinical studies are warranted to evaluate whether these efficacy and safety results prevail in less selected patients with localized PDAC treated with current neoadjuvant standard-of-care chemotherapy regimens.

## 6. Effects of Primary Thromboprophylaxis on Survival in Patients with PDAC

In addition to the effect on VTE incidence, the CONKO 004 and FRAGEM trials studied PFS and OS in patients with PDAC treated with LWMH thromboprophylaxis [96,97]. The CONKO 004 trial did not find a difference in the OS when comparing the enoxaparin to the observation arm; the median OS was 8.0 months in the observation arm and 8.5 months in the enoxaparin arm (HR = 1.01, 95% CI [0.87–1.34], *p* = 0.44). The median PFS did not differ, at 5.4 months in the observation arm and 5.0 months in the enoxaparin arm (HR = 1.06, 95% CI [0.84–1.32], *p* = 0.64). The FRAGEM trial also did not find an effect of thromboprophylaxis on PFS or OS. The median OS in the dalteparin group was 8.7 months, compared to 9.7 months in the control group (*p* = 0.682). In the dalteparin group, the median PFS was 5.3 months, compared to 5.5 months in the control group (*p* = 0.841). However, as the PFS and OS were not the primary outcome measures of these trials, these studies may not have had sufficient statistical power to investigate the effect of thromboprophylaxis on survival.

The post hoc analysis of the AVERT trial, which included patients with gastrointestinal cancer, compared the occurrence of all-cause death between the apixaban and placebo arms [101]. No difference in death occurrence was found between the apixaban and placebo group during the 180-day follow-up period, with death occurring in 30% of the patients in the apixaban group compared to 21% of the patients in the control group (HR = 1.73, 95% CI [0.97–3.09], *p* = 0.07). The prespecified subgroup analysis of the CASSINI study compared all-cause mortality in ambulatory patients with pancreatic cancer either treated with rivaroxaban thromboprophylaxis or placebo [106]. During the 180 days following randomization, there was no difference in deaths between the rivaroxaban and placebo group, there were 34 deaths (25%) in the rivaroxaban group and 33 (24%) in the placebo group (HR = 1.05, 95% CI [0.65–1.69], *p* = 0.85). However, these studies were not designed to study the effect of thromboprophylaxis on survival and may not have significant power.

A large individual patient data meta-analysis included 14 randomized controlled trials comparing LMWH with placebo or standard care in ambulatory patients with solid tumors [107]. A total of 823 patients with pancreatic cancer were included. This meta-analysis did not find an effect of LMWH on one-year mortality in PDAC patients (RR = 1.08; 95% CI [0.89–1.29]).

In conclusion, there is no evidence that thromboprophylaxis administration prolongs survival in patients with advanced PDAC treated with chemotherapy. The prospective and controlled trials were either small or underpowered to study the effect on survival. In addition, all trials included patients with locally advanced and metastatic PDAC. To establish whether thromboprophylaxis in the neoadjuvant setting influences survival, further clinical studies are required.

## 7. Current Guidelines on Thromboprophylaxis in PDAC

Currently, all guidelines either directly or indirectly recommend starting thromboprophylaxis in patients with PDAC treated with chemotherapy (Table 4) [23,24,88,108]. The European Society of Medical Oncology (ESMO) guideline recommends thromboprophylaxis in ambulatory patients with PDAC treated with chemotherapy, specifically LMWH at a higher dose [23]. The guidelines from the American Society of Clinical Oncology (ASCO), the International Initiative on Thrombosis and Cancer (ITAC), and the National Comprehensive Cancer Network (NCCN) recommend to start thromboprophylaxis in outpatients with cancer with an intermediate-to-high VTE risk, i.e., Khorona score ≥ 2, that are treated with chemotherapy [24,88,108]. As PDAC itself accounts for 2 points in the Khorana score, these guidelines indirectly advise thromboprophylaxis in all patients with PDAC undergoing chemotherapy. According to the latter guidelines, thromboprophylaxis with apixaban, rivaroxaban, and LMWH is suggested in patients without important risk factors for bleeding and drug–drug interactions [24,88,108].

A thromboprophylaxis duration of three months is suggested by the ESMO guideline, which is based on the three-month thromboprophylaxis period of the CONKO 004 and FRAGEM trials, making an extended administration non evidence-based [23]. The NCCN guideline advises to consider thromboprophylaxis for up to 6 months or longer in the case that the VTE risk persists [88].

According to the guidelines, thromboprophylaxis should be considered in all patients with PDAC treated with neoadjuvant chemotherapy. However, as these guidelines are based on current studies which generally focus on patients with locally advanced or metastatic PDAC, these recommendations should be considered with caution. As stated, additional clinical trials are needed to gain knowledge on the efficacy and safety of thromboprophylaxis in the neoadjuvant setting.

## 8. Postoperative Thromboprophylaxis in Patients with Resectable Pancreatic Cancer Undergoing Pancreatectomy

Cancer and major abdominal surgery are both risk factors for VTE. International guidelines on thromboprophylaxis following major abdominal surgery for malignancies are mainly based on two randomized controlled trials (RCTs). The FAME trial randomized 427 patients undergoing major abdominal surgery to either 7 or 28 days of LMWH (dalteparin). Among patients receiving a 28-day course of LMWH, the incidence of VTE was lower (7%) than in those receiving 7 days of LMWH (16%), while no increase in bleeding events was observed [109]. The placebo-controlled ENOXACAN II trial included 332 patients undergoing elective open surgery for abdominal or pelvic cancer, and was randomized between thromboprophylaxis for 6–10 days or an additional 21 days. This study also showed that an extended duration of enoxaparin prophylaxis was safe and reduced the incidence of VTE by 60% (95% CI [10,11,12,13,14,15,16,17,18,19,20,21,22,22,23,24,25,26,27,28,29,30,31,32,33,34,35,36,37,38,39,40,41,42,42,42,43,44,45,46,47,48,49,50,50,51,52,53,54,55,55,56,57,58,59,60,61,62,63,64,65,66,68,72,73,74,75,76,77,78,79,80,81]) [110]. Based on these studies, international guidelines recommend the extended use of postoperative VTE prophylaxis after abdominal or pelvic cancer surgeries for high-risk patients.

However, these studies and guidelines did not focus on patients undergoing resection for PDAC specifically. Since PDAC is a highly thrombogenic malignancy and pancreatic surgery is among the most extensive of major abdominal surgeries, these patients are considered to be at a very high risk of postoperative VTE. In addition to DVT and PE, these patients are also at an increased risk of postoperative thrombosis of the portomesenteric vein, especially if concomitant vascular resection and reconstruction are performed. Post-pancreatectomy thrombosis of the superior mesenteric vein (SMV) and portomesenteric vein (PV) occurs in 16.4–18% of patients, and varies according to the SMV/PV reconstruction technique [111,112]. However, these patients are also at a substantial risk of post-pancreatectomy hemorrhage (PPH), rendering the balance between thrombosis and bleeding fragile. PPH accounts for a significant portion of postoperative mortality [113,114,115]. Therefore, the risk–benefit ratio of extended thromboprophylaxis may be different for PDAC patients undergoing pancreatic surgery.

A few studies investigated thromboprophylaxis following pancreatic surgery for PDAC, and confirmed the efficacy of the general guidelines, which is that extended thromboprophylaxis was safe and did not increase the risk of PPH [83,116,117]. However, most studies were retrospective and varied in the VTE outcomes analyzed. For example, Sood et al. excluded portomesenteric thrombosis, Hayashi et al. included CVC-related thrombosis, and Imamura et al. only considered PE as VTE and did not document PPH. Regarding the optimal dose of LMWH prophylaxis, Hanna Sawires et al. retrospectively compared different doses of the LMWH dalteparin and demonstrated a two-fold higher rate of clinically relevant PPH when comparing a double dose (5700 units, once daily) to a single dose (2850 units, once daily) for six weeks postoperatively, while no benefit in VTE was found [118]. The only prospective study carried out was a single-arm cohort study by Hashimoto et al., which analyzed 103 patients who underwent pancreatic resection and received postoperative enoxaparin (2000 IE, twice daily) for 14 days or until discharge [119]. In this study, no patients developed symptomatic VTE, two patients developed asymptomatic VTE (2%), and three patients developed PPH (3%). When the authors compared their results to the literature, they concluded that enoxaparin may prevent VTE without increasing the risk of PPH. Because of the scarcity of high-evidence studies focusing on thromboprophylaxis following pancreatic surgery for PDAC, no guidelines provide a clear recommendation on this topic.

During pancreatic surgery for PDAC, vascular resection and reconstruction can be required for a radical resection, increasing the risk of PVT. A survey among 167 surgeons by Groen et al. identified a wide variation in the use of thromboprophylaxis (>10 regimens) and reported that 39% of surgeons adjust thromboprophylaxis following venous resection [120]. Recently, the Americas Hepato-Pancreato-Biliary Association (AHPBA) guidelines were published, in which a systematic review was included focused on postoperative anticoagulation after pancreatic resection with vascular reconstruction [121]. This systematic review compared 27 retrospective studies on the use of anticoagulation to ten retrospective studies which did not use postoperative anticoagulation [121]. They reported a significant heterogeneity across the anticoagulation policies, and the types of vascular reconstruction significantly differed between the two groups. Interestingly, the incidence of VTE within 30 days was higher in the anticoagulation group when compared to no anticoagulation (5% vs. 2%, *p* = 0.009). No difference was found in PPH rates (7% vs. 7%, *p* = 0.86). This review concluded that it was impossible to derive a firm conclusion regarding anticoagulation in this setting based on the available data.

In summary, the prophylaxis of VTE following pancreatic resection for PDAC requires a tailored approach, especially considering the risk of PPH. Considering the collective current literature, extended thromboprophylaxis with a prophylactic dose of LMWH for four-to-six weeks following pancreatic resection is advised to prevent postoperative VTE. However, further research is required with a focus on vascular resections.

## 9. Contraindications to Primary Thromboprophylaxis

The FRAGEM, CONKO 004, AVERT, and CASSINI trials analyzed thromboprophylaxis in a selected patient group [96,97,100,102,122]. Table 5 provides a detailed overview of the exclusion criteria of these studies. Patients with a high bleeding risk, low performance score (Karnofsky performance status < 60 or ECOG performance status ≥ 3), thrombocytopenia (platelet counts below 50 × 10^9^/L), and impaired renal function (generally creatinine clearance < 30 mL/min) were excluded. In the FRAGEM trial, patients on antiplatelet treatment and patients with central venous access devices were also excluded. As effectivity and toxicity profiles remain unclarified for patients with these characteristics, thromboprophylaxis should not be administered to these patients. The final decision on starting thromboprophylaxis needs tailoring to the individual patient and the risk of thrombosis should be weighed against the risk of bleeding.

### 9.1. Drug–Drug Interactions of Chemotherapy and Thromboprophylaxis

Patients with cancer more often experience drug–drug interactions (DDIs) due to polypharmacy and absorption problems resulting from chemotherapy-induced vomiting, diarrhea, and liver dysfunction, leading to unstable drug levels. LMWHs are known for their predictable pharmacokinetics and minimal DDIs [123]. DOACs require more consideration of DDIs, especially with drugs that compete for or inhibit the P-glycoprotein and CYP3A4-type cytochrome P450 [124,125]. For platinum-based agents (e.g., oxaliplatin), topoisomerase inhibitors (e.g., irinotecan) and pyrimidine analogues (e.g., 5-FU, gemcitabine), commonly used chemotherapeutics in the neoadjuvant treatment of PDAC, no relevant DDIs are anticipated with DOACs [126]. For the coadministration of paclitaxel with apixaban or rivaroxaban, caution is required due to the possibly reduced DOAC plasma levels [126].

Anti-emetic prophylaxis is often prescribed for chemotherapy-induced nausea and vomiting. Neurokinin antagonists, serotonin-5-HT3-antagonist, dopamine antagonists, and benzodiazepines do not interact with DOACs. Dexamethasone and prednisone are advised to be co-administered with caution, as they could lower apixaban and rivaroxaban levels [126]. Apixaban and rivaroxaban are mainly metabolized by CYP3A4. As dexamethasone and prednisone are inducers of both CYP3A4 and P-glycoprotein, the clearance is increased, and plasma levels are potentially reduced [127]. Granulocyte colony-stimulating factors, administered to reduce the severity and duration of chemotherapy-induced neutropenia, do not interact with DOACs [87].

In summary, commonly administered chemotherapeutics and chemotherapy-supporting drugs in the neoadjuvant treatment of patients with PDAC do not have severe DDIs with LMWH and DOACs, and are thus permitted.

### 9.2. Interruption of Primary Thromboprophylaxis during Neoadjuvant Chemotherapy

Interruption of anticoagulation treatment is more often required in patients with cancer due to a changed bleeding risk or invasive procedures during the course of treatment. Chemotherapy can cause adverse effects such as thrombocytopenia and mucositis, increasing the risk of bleeding [57,128]. Tumor progression can also lead to an increased bleeding risk, for example, via gastroduodenal tumor invasion or PDAC-related portal hypertension [129,130,131].

#### Chemotherapy-Induced Thrombocytopenia

Chemotherapy-induced thrombocytopenia is a common complication in PDAC patients. PDAC patients are often treated with either gemcitabine-based chemo(radio)therapy or FOLFIRINOX chemotherapy in the neoadjuvant setting. Grade-3 or -4 thrombocytopenia incidences of 9% have been reported for FOLFIRINOX [57], 4–9% for gemcitabine [57,58], and 4–13% for gemcitabine in combination with nab-paclitaxel [58,132] in patients with locally advanced or metastatic PDAC.

The European Hematology Association (EHA) recently published a guideline on the management of antithrombotic treatments in thrombocytopenic patients with cancer [133]. For grade-1–2 thrombocytopenia (TP) (platelet count 50–100 × 10^9^/L), a standard prophylactic dose of LMWH, apixaban, and rivaroxaban can be used. In the case of a grade-3 TP (platelet count 25–50 × 10^9^/L), DOACs are contraindicated and a standard prophylactic dose of LMWH may be considered in the absence of additional bleeding risk factors, and if platelet counts are either stable or monitored closely. For grade-4 TP, the guideline recommends interrupting any prophylactic and therapeutic anticoagulant treatment, for both DOACs and LMWH. Restarting antithrombotic therapy can be considered once the platelet count is consistently above the threshold deemed for antithrombotic medication, being >50 × 10^9^/L. Importantly, proton pump inhibitors (PPIs) are advised in all cancer patients as single or combined antithrombotic drugs, to prevent GI bleeding.

## 10. Conclusions and Future Directions

Historically, patients with PDAC eligible for tumor resection were offered immediate surgery. Since recent studies showed a survival benefit in patients treated with neoadjuvant chemo(radio)therapy compared to immediate surgery, more and more ambulatory patients are treated with chemotherapy in the neoadjuvant setting.

It is well-known that patients with PDAC are at a high risk of developing thrombosis. However, since neoadjuvant treatment for PDAC has only recently become standard of care, data are scarce on VTE incidence and risk factors for VTE development during this period. Studies have shown that VTE in PDAC patients is associated with a decreased survival and a lower rate of completion of the intended treatment plan.

Although current guidelines recommend primary thromboprophylaxis for PDAC patients receiving systemic treatment, data on such an approach in patients receiving neoadjuvant treatment are lacking. Moreover, there are no risk assessment tools or biomarkers that can identify PDAC patients at the highest risk of VTE. For example, the widely used Khorana score does not seem to be able to distinguish between PDAC patients with an intermediate and high VTE risk. Identifying high-risk PDAC patients receiving neoadjuvant treatment would be helpful in selecting the patients that are likely to benefit the most from primary thromboprophylaxis.

Patients with PDAC undergoing pancreatic surgery are not only at a risk of postoperative thrombosis, but also PPH. Despite their risk of PPH, extended thromboprophylaxis with prophylactic LMWH for four-to six-weeks has been shown to be effective and safe in the prevention of VTE, and is hence recommended.

Further studies are required to determine the effects of perioperative primary thromboprophylaxis in patients with PDAC treated with neoadjuvant chemotherapy.

## Figures and Tables

**Table 1 cancers-15-03546-t001:** Incidence of VTE in patients with localized PDAC (treated with neoadjuvant chemotherapy).

Study	Study Size	Cancer Stage, *n* (%)	Chemotherapy, *n* (%)	VTE Incidence • Stage, *n* (%)	VTE incidence • Chemotherapy, *n* (%)
*Prospective*					
Frere et al., 2020 [22]	731	RPC: 208 (29.0) BRPC: 105 (14.6) LAPC: 212 (26.9)	-	Total: 97 (19) • RPC: 31 (21) • BRPC: 17 (11) • LAPC: 49 (33)	-
Krepline et al., 2016 [27]	260	RPC: 109 (42) BRPC: 151 (58)	5-FU: 98 (37) Gemcitabine: 84 (32) Platinum agent: 110 (42)	Total: 26 (10) • RPC: 9 (8) • BRPC: 17 (11)	• 5-FU: 13/98 (13) • Gemcitabine: 5/84 (6) • Platinum agent: 13/110 12)
Walma et al., 2021 [28]	326	LAPC: 326 (100)	FOLFIRINOX: 252 (77) Nab-paclitaxel/gemcitabine: 33 (10) Gemcitabine: 41 (13)	Total: 20/326 (6)	• FOLFIRINOX: 17/252 (7) • Nab-paclitaxel/ gemcitabine: 2/33 (6) • Gemcitabine: 1/41 (2)
Katz et al., 2016 [29]	22	BRPC: 22 (100)	mFOLFIRINOX: 22/22 (100)	Total: 3 (14)	• mFOLFIRINOX: 3/22 (14)
*Retrospective*					
Barreau et al., 2021 [30]	174	LAPC: 56 (32)	-	Total: 46 (26) LAPC: 13 (23)	
Tahara et al., 2018 [31]	27	LAPC: 21 (78) Metastatic: 6 (22)	FOLFIRINOX: 10 (37) Nab-paclitaxel/gemcitabine: 11 (41)	Total: 6/27 (22) ^a^	• FOLFIRINOX: 5 (42) • Nab-paclitaxel/gemcitabine 1 (7)
Hanna-Sawires et al., 2021 [32]	361	I: 62 (17) II: 152 (42) III: 61 (17)	FOLFIRINOX: 6 Gemcitabine/radiotherapy: 11	Total: 64/361 (18) I: 7/62 (11) II: 24/152 (38) III: 9/61 (14) During neoadjuvant therapy: 2 (3)	

^a^ VTE incidence for both locally advanced and metastatic PDAC, VTE incidence for only LAPC was not reported in study. BRPC: borderline resectable pancreatic cancer; LAPC: locally advanced pancreatic cancer; mFOLFIRINOX: modified FOLFIRINOX; RPC: resectable pancreatic cancer; VTE: venous thromboembolic event.

**Table 2 cancers-15-03546-t002:** Predictors included in the VTE risk assessment tools for cancer patients.

Predictors	Khorana Score [67]	Vienna Model [68]	ONCOTHROMB [69]	Li Model [70]	ONKOTEV [71]
**Cancer characteristics**					
Cancer type	**X**	**X**	**X**	**X**	**X**
Cancer stage			**X**	**X**	**X**
Vascular/lymphatic compression					**X**
**Patient characteristics**					
Previous VTE				**X**	**X**
Immobilization				**X**	
Recent hospitalization >3 days				**X**	
BMI	**X**		**X**	**X**	**X**
Asian/Pacific islander				**X**	
**Biomarkers**					
Hemoglobin or use of RBC growth factors	**X**			**X**	**X**
Leukocytes	**X**			**X**	**X**
Platelet count	**X**			**X**	**X**
D-dimer		**X**			
Soluble P-selectin					
Genetic germline mutations			**X**		

BMI: body mass index; RBC: red blood cell.

**Table 4 cancers-15-03546-t004:** Current guideline recommendations on primary thromboprophylaxis in ambulatory patients with PDAC treated with chemotherapy.

Guideline	Recommendation	Contraindications	Duration	Thromboprophylaxis Regimen
ASCO (2023) [108]	Start thromboprophylaxis in outpatients with a Khorana score ≥ 2, starting a new chemotherapy regimen	• Bleeding risk • Drug–drug interaction	-	Apixaban (2.5 mg PO twice daily), rivaroxaban (10 mg PO once daily) or LMWH
ESMO (2023) [23]	Start thromboprophylaxis in ambulatory PDAC patients on first-line anticancer treatment	-	Maximum of 3 months	LMWH at higher dose: 150/IU/kg dalteparin or 1 mg/kg enoxaparin
ITAC (2022) [24]	• Start thromboprophylaxis in ambulatory patients with a Khorana score ≥ 2, receiving anticancer therapy • Start thromboprophylaxis in ambulatory patients with locally advanced or metastatic PDAC treated with anticancer therapy	• High bleeding risk • Active bleeding	-	Apixaban (2.5 mg PO twice daily), rivaroxaban (10 mg PO once daily) or LMWH
NCCN (2021) [88]	Start thromboprophylaxis in outpatients with a Khorana score ≥ 2, receiving/starting chemotherapy	• Active bleeding • Thrombocytopenia (platelet count < 50,000/µL) • Hemorrhagic coagulopathy or known bleeding disorder in the absence of replacement therapy • Indwelling neuraxial catheters • Neuraxial anesthesia/lumbar puncture • Interventional spine and pain procedures	Up to 6 months or longer, if the risk persists	• Patients with Khorona score ≥ 2: apixaban (2.5 mg PO twice daily), rivaroxaban (10 mg PO once daily) or LMWH • Patients with advanced unresectable or metastatic PDAC: dalteparin 200 IU/kg SC daily 1 month, then 150 IU/kg SC daily 2 months or enoxaparin 1 mg/kg daily 3 months, then 40 mg SC daily ^a^

^a^ Data support the administration of dalteparin and enoxaparin for patients with advanced unresectable and metastatic pancreatic cancer. ASCO: American Society for Clinical Oncology; ESMO: European Society for Medical Oncology; ITAC: International Initiative on Cancer and Thrombosis; IU: international unit; LMWH: low-molecular-weight heparin; PDAC: pancreatic ductal adenocarcinoma; PO: per os; SC: subcutaneously.

**Table 5 cancers-15-03546-t005:** Exclusion criteria for trials investigating thromboprophylaxis in PDAC patients.

	FRAGEM Trial [96]	CONKO 004 Trial [97]	CASSINI Trial [122]	AVERT Trial [101]
*Thromboprophylaxis*	Dalteparin, 200 IU/kg, once daily	Enoxaparin, 1 mg/kg, once daily	Rivaroxaban, 10 mg, once daily	Apixaban, 2.5 mg, twice daily
*Exclusion criteria*	Karnofsky performance status < 60	Karnofsky performance status < 60	ECOG performance status ≥ 3	
		Body weight < 45 kg or > 100 kg		Body weight < 40 kg
	CrCl < 50 mL/min	CrCl < 30 mL/min	CrCl < 30 mL/min	CrCl < 30 mL/min
	• Platelets < 100 × 10^9^/L • Absolute neutrophil count < 2 × 10^9^/L • White cell count < 3 × 10^9^/L • INR > 1.5 • Adequate liver function Bilirubin > 1.5 upper limit of normal	• Platelets <100 × 10^9^/L • White cell count < 3.5 × 10^9^/L		• Platelets < 50 × 10^9^/L
	• Obvious contraindication to anticoagulation	• Major hemorrhage within the last 2 weeks • Severely impaired coagulation • Active gastrointestinal ulcers • Major surgery within the last 2 weeks	• Bleeding diathesis • Hemorrhagic lesions • Active bleeding • Conditions with a high risk of bleeding	• Increased risk of significant bleeding • Hepatic disease with coagulopathy
	• Anticoagulation treatment • Antiplatelet treatment (i.e., Aspirin > 75 mg, clopidogrel, etc.)	• Anticoagulation treatment		• Anticoagulation treatment • Medication contraindicated with apixaban

CrCl: creatinine clearance; INR: international normalized ratio; IU/kg: international units/kilogram.

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
