# Peer review of "Venous Thromboembolism and Primary Thromboprophylaxis in Perioperative Pancreatic Cancer Care"

_cancers, 2023, doi:10.3390/cancers15143546_

Round 1

Reviewer 1 Report

Thank you for giving me the opportunity to peer review. The authors have reviewed the risk of developing VTE and prevention of VTE in pancreatic cancer patients. I believe that this paper will provide very useful information for clinicians.

In discussion section, a platelet count lower than 50 to 100 x 109 platelets are described contraindications.Is it correct that 50 to 100 x 109 are described contraindications. ?

 In discussion section, Tumor growth can also lead to an increased bleeding risk, by example by gastroduodenal tumor invasion~Is it correct that Tumor growth can also lead to an increased bleeding risk, for example by gastroduodenal tumor invasion~?

Author Response

Dear reviewer,

We would like to thank the editor and the reviewers for taking the necessary time and effort to review our manuscript. We appreciate the valuable comments and suggestions, which helped us in improving our manuscript.

We have addressed all of the comments and concerns in a point-by-point response.

Thank you for giving me the opportunity to peer review. The authors have reviewed the risk of developing VTE and prevention of VTE in pancreatic cancer patients. I believe that this paper will provide very useful information for clinicians.

Comment 1: A platelet count lower than 50 to 100 x 109 platelets are described contraindications.

→Is it correct that 50 to 100 x 109 are described contraindications?

Thank you for your complement.

Our review has been shortened during the revision round. Therefore, section 9.2 on contraindications for primary prophylaxis, which your comment refers to, has been edited and the content has been included in other sections.

In table 5, we now summarize the exclusion criteria for the trials investigating thromboprophylaxis in patients with advanced pancreatic cancer. The FRAGEM trial and CONKO 004 trial excluded patients with a platelet count lower than 100 x 109/L. The CASSINI and AVERT trials, investigating DOACs, excluded patients with a platelet count below 50 x 109/L. In section 9 we conclude that for patients with platelets below 50 x 10 x 109/L effectivity and safety profiles remain unclarified and that therefore thromboprophylaxis should not be administered to these patients. In addition, the NCCN guideline proposes thrombocytopenia with a platelet count below 50 as a contraindication, which is reported in table 4 of our manuscript.

We hope that this section addresses your concern adequately.

Comment 2: Tumor growth can also lead to an increased bleeding risk, by example by gastroduodenal tumor invasion→ Is it correct that Tumor growth can also lead to an increased bleeding risk, for example by gastroduodenal tumor invasion?

In our manuscript we indeed mention that tumor growth can lead to bleeding by gastroduodenal tumor invasion. We refer to a paper by Munoz et al. (2020). In this paper, data was collected on non-postoperative gastrointestinal bleeding in patients with pancreatic ductal adenocarcinoma. They included 72 patients that had 94 episodes of gastrointestinal bleeding, the main cause of gastrointestinal bleeding was gastroduodenal tumor invasion (56.4%) and esophageal bleeding due to left-sided portal hypertension due to tumor obstruction (19.1%). We find the results of this study very interesting and hence mentioned this phenomenon in our manuscript. To add to the evidence on this topic, we have added another paper by Wang et al. that retrospectively reviews 246 cases of gastrointestinal bleeding associated with pancreatic cancer.

We hope that the elaboration on this study and the addition of another paper provides sufficient evidence that pancreatic tumor growth can lead to gastrointestinal bleeding.

We look forward to hearing from you regarding our submission and to respond to any further questions and comments you may have.

Sincerely yours,

Judith de Vos-Geelen

Reviewer 2 Report

The review about venous thromboembolism and primary thromboprophylaxis in perioperative pancreatic cancer care is an detailed anaylysis of the current knowledge about this topic, especially in patients during neoadjuvant chmotherapy. 

Therefore, studies and sufficient data to provide evidence based recommendations and guideline are not sufficiently available. 

1. However, the mansucript is much too long and short be substantially shortened (please shorten about 1 third of the text).  It is not necessary to mention every study in detail within the text; most studies are then shown in the tables and listed in the references. Just give a summary of the knowledge provided by the studies for every subgroup of patients. 

2. Please mention also shortly the perioperative risk of portal vein thrombosis in patients with pancreatic cancer and those under neodajuvant chemotherapy. 

minor corrections required 

Author Response

Dear reviewer,

We would like to thank the editor and the reviewers for taking the necessary time and effort to review our manuscript. We appreciate the valuable comments and suggestions, which helped us in improving our manuscript.

We have addressed all of the comments and concerns in a point-by-point response.

The review about venous thromboembolism and primary thromboprophylaxis in perioperative pancreatic cancer care is an detailed anaylysis of the current knowledge about this topic, especially in patients during neoadjuvant chmotherapy.

Therefore, studies and sufficient data to provide evidence based recommendations and guideline are not sufficiently available

1. However, the mansucript is much too long and short be substantially shortened (please shorten about 1 third of the text). It is not necessary to mention every study in detail within the text; most studies are then shown in the tables and listed in the references. Just give a summary of the knowledge provided by the studies for every subgroup of patients.

Thank you for your compliments.

We took the reviewer’s advice to shorten our review, especially the sections that were also described in the tables. The main text is reduced from 7028 to 5212 words. We hope this more concise manuscript will improve the readability of our review.

 2. Please mention also shortly the perioperative risk of portal vein thrombosis in patients with pancreatic cancer and those under neodajuvant chemotherapy.

We would like to thank the reviewer for this suggestion. We have elaborated on the risk of portal vein thrombosis in section 8. We have further investigated the evidence on portal vein thrombosis (PVT) in the neoadjuvant setting. Very little evidence is available specifically on PVT during neoadjuvant treatment. One paper by Krepline et al. mentioned one out of 260 patients undergoing neoadjuvant chemotherapy developed PVT. Therefore, we chose not to specifically mention the preoperative PVT risk in our manuscript.

We look forward to hearing from you regarding our submission and to respond to any further questions and comments you may have.

Sincerely yours,

Judith de Vos-Geelen

Round 2

Reviewer 2 Report

the authors have shortened and improved their manuscript now acceptable for publication. 

minor corrections